# "My Thighs Can Squash You": Young Māori and Pasifika Wāhine Celebration of Strong Brown Bodies

Mihi Joy Nemani [1,2,*] and Holly Thorpe [2,*]

1    School of Sport, Manukau Institute of Technology/Te Pūkenga, Auckland 2023, New Zealand
2    Te Huataki Waiora-School of Health, University of Waikato, Waikato 3240, New Zealand
*    Correspondence: mihi.nemani@waikato.ac.nz (M.J.N.); holly.thorpe@waikato.ac.nz (H.T.)

**Abstract:** Media representations and dominant social constructions of the 'ideal' physique for young women are often framed through a Westernised lens that focuses on heteronormative, White able-bodied aesthetics of beauty and femininity. Until very recently, the imagery available for young women to connect with and aspire to has been highly limited, failing to represent the embodied cultural beliefs that Indigenous and culturally-minoritised young women may have towards the gendered body. In this paper, we draw upon focus groups (wānanga) and digital diaries with young, physically active Māori and Pasifika wāhine (women) in Aotearoa New Zealand, to reveal how they are making meaning out of dominant framings of beauty, and drawing upon cultural knowledge to refuse such portrayals, instead reclaiming power in their own bodies. Working at the intersection of Mana Wahine and Masi methodologies, this article amplifies the voices of young Māori and Pasifika wāhine who actively participate in sport and/or physical activity, embrace and appreciate their strong brown bodies, and are critically reading and rejecting dominant Western framings of beauty and femininity. In so doing, this paper contributes to a growing international dialogue about the need for new culturally-informed understandings of body image by young women from Indigenous and culturally marginalised communities.

**Keywords:** young women; body image; feminist Indigenous methodologies; Mana Wahine; Masi methodology; Aotearoa New Zealand

## 1. Introduction

Over recent decades, scholars around the world have examined the powerful role of the media in reproducing unrealistic feminine aesthetic ideals [1–5]. While the ideal body shapes and types have changed over time, Westernised views of women's bodies continue to influence socially accepted norms that often disregard, or are contrary to, the views and perspectives of ethnic minority or 'othered' bodies [6–9]. With the significantly increased use of social media platforms (Facebook, Instagram, TikTok), a greater diversity of visual representations of ethnic minority and traditionally othered women's bodies (i.e., fat, disabled) are available [10,11]. Yet, across many media outlets, the dominant narrative continues to focus on an ideal physique (i.e., thin, toned, young) related to White heteronormative standards. Numerous scholars have conducted research on young women and body image [12–16], powerfully providing evidence of the impact of such media portrayals on young women's bodily dissatisfaction, with a heightened risk of disordered eating practises and mental health issues. To date, only a handful of scholars have focused on the ways in which young women from Indigenous and ethnic minority communities make meaning of such imagery [17,18].

In this paper, we draw upon wānanga (focus groups) and digital diaries to examine how young Māori and Pasifika wāhine (wahine is the Māori word for woman, and wāhine (hyphenated) is the Māori word for women) (16 to 25 years) who live in Aotearoa (New Zealand) understand body image. Drawing on Indigenous feminist methodologies Mana

Wahine and Masi methodology, and an intersectional approach, we report the findings for this paper under two main themes: (1) influences on body image, and (2) navigating body-image stereotypes. As we amplify the voices of the young wāhine, this article illustrates how they navigate Westernised views, and use their cultural knowledge of healthy bodies to powerfully reclaim what their bodies represent, and how they believe their bodies should be viewed.

**2. Literature: Context, Positioning, and Theoretical Framing**

Many academics have contributed to researching the complexities related to body image, and have provided insights, perspectives, and knowledge on the nuances of this topic [19–22]. This paper takes a socio-cultural approach, drawing upon the intersections of gender and ethnicity in relation to body image, with a particular focus on body positivity. In this section, we focus on the literature that examines body image in relation to the experiences of women from Indigenous, ethnic-minority, or 'othered' communities. We begin with an examination of the literature on intersectionality [23,24], to understand the nuanced experiences of women at the intersection of gender and ethnicity, then present the literature on body image, based on work conducted with Black women in America, Aboriginal women in Canada and, finally, young Māori and Pasifika women in Aotearoa.

Intersectionality is a theory created and developed by Kimberlé Crenshaw [24,25], a Black African-American civil rights advocate and Professor of Law, who saw the need for a theory that acknowledged Black women and their nuanced lived experiences. Using the intersection of gender and race, Crenshaw [23,24] centred the voices of Black women and, through their lived experiences, began to unveil the systems and structures that discriminated against them in policy and practice. Intersectionality has been used by many scholars who understand the complexities of identity for Indigenous, ethnic minority, and othered women, by revealing the structures of power and privilege that create their multiple minoritised positions in Westernised societies [26–28]. We echo the use of intersectionality by Watson, Lewis, and Moody [29], who explain that it "provides a more nuanced understanding of the ways that the history of Black women's gender and racial oppression influence their specific body image concerns" [29] (p. 2). This paper contributes to scholarship on intersectionality by providing knowledge from young Māori and Pasifika wāhine about their body-image experiences in Aotearoa, New Zealand.

Recent literature on body image is showing a shift away from slim ideals, and towards more voluptuous bodies, or the 'slim-thick' ideal that includes having a slim waist, large breasts, and large buttocks [30,31]. Some literature identifies the turn towards more curvaceous ideals as heavily influenced by Black and Hispanic cultures, and particularly celebrities such as the Kardashian women, Jennifer Lopez, Beyoncé, and Serena Williams, among others [30,32]. While these ideals have resonated with Black, Brown, and White women around the world, the new ideal to be both fat-free, toned, and curvaceous presents another largely unrealistic ideal [33]. Other literature is also showing a shift from slim to strong, wherein the slim, muscular, fit body is also being signalled as an ideal body [31,34,35]. Although the ideals may have changed from slim to slim-thick, or slim to slim-strong, they are still unrealistic and unattainable for many women.

A handful of scholars have conducted body-image research on women from Indigenous, ethnic-minoritised, and othered communities, and have identified nuanced cultural understandings that differ from White Westernised views [8,17,29,36–40]. For example, body-image research on Black women in America has highlighted a need for culturally informed methods and measures that consider their unique body-image concerns [40]. Lowy et al. [40] explain that, in addition to the influences of White standards of ideal body image, they also face cultural pressures that relate to Black women's standards of body image relating to hair style, skin colour, and curviness. Similarly, Viladrich and colleagues [41] found Hispanic women living in the US experiencing a paradoxical body image, between the thin-fit White ideal and the "Latina curvy shape as its counter-image" [41] (p. 20). Importantly, scholars recommend that research which enables culturally-specific ideas and

knowledge of body image is necessary, as it has the potential to show the agency of Black, Brown, and Indigenous women and their lived experiences, despite the pervasiveness of White, Western ideals and the social pressures that they navigate [40,42]. While some scholars have focused on the ways in which women navigate conflicting body-image ideals between their own cultures and the dominance of White, Westernised versions of femininity and beauty, others recommend taking strengths-focused approaches, wherein body positivity can emphasise the ways that Black, Brown, and Indigenous women appreciate and embrace the uniqueness of their bodies [29].

Historically the research on body image has focused on body-image dissatisfaction [43–46]. However, recently, there has been a growing body of scholarship focused on body positivity, or having a holistic positive evaluation of your body [47]. Scholars have documented how the 'body positive' movement that proliferated via social media has been shaped by postfeminist discourses, and is connected to neoliberal capitalist values (i.e., consumerism, empowerment, 'love thy self') [48,49]. As Sastre [48] found, online body positivity discourse "more closely mirrors than challenges a neoliberal paradigm of bodily compliance" (p. 929). Despite this, a number of scholars argue that conducting research on participants with body-positive perspectives, and understanding the diverse sources of such positivity (i.e., social media, families, peers, cultural knowledge, religion), can provide a set of unique insights relating to body image that have the potential to provide tools for practitioners working in body-image-related fields [47,50–53]. Some of these characteristics include having a higher self-esteem, having a broader view on beauty ideals that go beyond physical appearance, rejecting and criticising media representations of an unrealistic body image, and having an appreciation of the functionality of their body, and a desire to practise self-care and compassion [50,51,54].

Leboeuf [55] extends the concept of body positivity, and explains that, while it is an important factor to help women embrace the unique features of their bodies, "body pride marks a step toward overcoming unwarranted shame about one's own body and valuing all body types, especially stigmatized ones" (p. 122). Body pride is a concept that has been used across an array of contexts, including among the disability, queer, and Fat communities. As Leboeuf [55] writes, "body positivity should be understood as a response to limiting body shame that can be achieved through the cultivation of proper body pride" (p. 124). Body pride has also been used to understand the body-image ideals of Indigenous, ethnic-minority, and culturally diverse groups who are influenced by Westernised body-image ideals [54]. Such work starts from culturally-informed ways of knowing the body that lead to distinctive forms of body pride and positivity, shaped less by popular social media discourse, and more by local communities and ways of understanding health and wellbeing. Thus, coupled with intersectionality, body pride has the potential to provide a framework which can be used to empower and amplify the voices of young Indigenous, ethnic-minority, and culturally diverse women.

The work of Coppola and colleagues offers a powerful example of scholars using a body-pride approach to amplify the voices of young Indigenous people [36,56,57]. One of the studies focused solely on young Aboriginal women in Canada, where the participants explained how body pride created a space for their cultural understandings of body image [36]. The young Aboriginal women in this study linked their cultural identity to body pride, and how they embraced all aspects of their bodies, which included embodied cultural practices, such as song and dance. While they acknowledged the negative body-image stereotypes associated with Aboriginal youth, they argued that body pride was more important, as it was a direct influence of their cultural identity. For them, body pride meant having resolve in their identity, which included "Being proud of who you are, [and] not being ashamed of your background" [36] (p. 324). Interestingly, the young Aboriginal women also placed the responsibility on their elders to do something to change the way in which Aboriginal people are viewed in society, and suggested that this would also help increase body pride for other young Aboriginal women [36]. Here, we see young Indigenous women using their knowledge of body image in ways that are culturally specific

for them, while placing the responsibility for facilitating positive narratives for their culture in broader society on their elders.

In Aotearoa, a few scholars have conducted research on body image specifically for Māori wāhine [58,59] and Pasifika wāhine [60,61]. Some of the themes from this work include recognising that Māori and Pasifika people generally have bigger bodies, but lower levels of 'body dissatisfaction', compared to other ethnic groups, and that they have an appreciation for how their bodies function [59–63]. While this research has provided some important cultural understandings on body image for young wāhine, over the past decade or so since these studies were conducted, there has been a significant increase in societal influences, specifically with social media, which indicates a need for more current research. There is also potential for fresh perspectives on the body-image experiences of young wāhine in Aotearoa from a mixed ethnic background. These are both areas that this research aims to address through an intersectional lens, using culturally-responsive body positivity and body pride.

A handful of scholars have conducted body positivity research on young women in Aotearoa, with the majority of the samples consisting of NZ European University students. To date, none of this research has focused specifically on the experiences of young Māori, Pasifika, or culturally-othered (we use 'culturally othered' to acknowledge young wāhine who are part of groups that share similar beliefs and experiences, such as religion or disability) wāhine [47,64,65]. Acknowledging the gap in their research, and echoing the findings of other international academics [17,38,66], Poulter and Treharne [47] highlight the need for further research to be conducted that will help with cultural and religious understandings of body image for Indigenous, ethnic minority, and othered young women.

Herein, this paper aims to contribute to the work conducted by academics, internationally and in Aotearoa, who have identified the need to understand and draw upon knowledge of body image for young women from Indigenous, ethnic minority, and othered communities. We do this by focusing on both young Māori and Pasifika wāhine, and use intersectionality, with culturally learned body positivity and body pride, to unveil the ethnic and gendered experiences that frame their lived realities.

### 2.1. Context: Māori and Pasifika Connections in Aotearoa

Māori are the Indigenous people of Aotearoa who were part of the Polynesian migration, around AD 1280 [67]. Over 500 years after Māori first landed on and claimed Aotearoa as their homeland, the first White man arrived, marking the beginning of colonisation by the British Crown (hereafter, the Crown) [68]. In an attempt to maintain peace between Māori and the Crown, Te Tiriti o Waitangi (the Treaty of Waitangi) was signed, and is a living agreement between both parties. As part of this responsibility, the Crown agreed to enhance, preserve, and protect all things Māori, including the people, language, land, and culture [68]. This also means that all government agencies in Aotearoa are required to include and embed Te Ao Māori (Māori worldviews) in policy and practice.

The history of Pacific People migrating to Aotearoa is rich and diverse, as it involves the stories of people from many different countries, including Samoa, Tonga, Fiji, Cook Islands, and Niue [69]. Currently, the majority of Pacific People living in Aotearoa were born in Aotearoa, and have lived here all their lives. They are also second or third generation to their parents or grandparents who initially migrated here, so their lived experiences are very different to those individuals who were born and raised in the Islands they are Indigenous to. Navigating their cultural identities can often be challenging, particularly if they are raised in a more Westernised or mixed-ethnic family environment [70,71]. It is important to acknowledge that the term 'Pasifika' is heavily contested with many Pacific scholars, due to it not addressing the nuances and richness from each ethnic group. It also tends to exclude some Pacific Island peoples, such as those from the Solomon Islands, Kiribati, and Vanuatu [71–73]. Fa'avae [74], unpacks the use of the words Pasifika and Pacific, by grounding them in the identity that they represent, as well as their respective whakapapa (genealogy and ancestral lineage). He explains that the word 'Pasifika' relates

to Pacific Island people who whakapapa from Samoa, Tonga, Fiji, Niue, Tokelau, and the Cook Islands, as well as Pacific Island people who were born/or who have lived most of their life in Aotearoa [74]. Resonating with Fa'avae [74], we use Pasifika throughout this article as it represents the identity of the young Pasifika wāhine in our research.

There are important connections between Māori and Pasifika people, including historical ones, through Polynesian migration [75–77], and cultural ones, through language and identity [78]. They have also experienced similar social experiences relating to health, education, lower socio-economic living conditions, inter-cultural partnerships, and negative media representations [79–86]. Māori and Pasifika people have also often been grouped together in research, policy, and media, due to cultural similarities, and the disparities they experience in society compared to other ethnic groups [87–90]. Lastly, connections also exist among Māori and Pasifika who have a mixed-ethnic heritage, which in turn provides nuanced lived experiences that are gaining recognition in academia [91,92].

### 2.2. Cultural Context: Body Image and Hauora in Aotearoa

Indigenous scholars of Aotearoa focusing on young people's (rangatahi) health and wellbeing show that having a strong connection to their culture is highly important for better mental health [93,94]. These researchers have found that cultural identity and Indigenous ways of knowing health and wellbeing are important for supporting youth in a context where racial discrimination, inequities, and injustice continue to impact their everyday lives. In particular, Whare Tapa Wha (an Aotearoa health model) has been shown to impact and influence their engagement with health and wellbeing [95]. Whare tapa wha uses the analogy of a wharenui (a Māori meeting house that forms part of a Marae, which are meeting grounds that have a spiritual and cultural significance for Māori people; each Marae belongs to a specific Māori tribe or sub-tribe (Moeke-Pickering 1996)) which is upheld by four strong pillars. These four pillars include, "te taha tinana (physical wellbeing) ... te taha hinengaro (mental and emotional wellbeing) . . .te taha whānau (extended family and social wellbeing) . . .. [and] te taha wairua (spiritual wellbeing)" [96] (p. 61). For total wellbeing, all four pillars need to be attended to, strengthened, and balanced, much like the pillars of a wharenui [95–97].

The Whare Tapa Wha model is taught in schools as a critical part of the health curriculum [95], and it also shapes the everyday experiences of rangatahi (young people) with whānau (immediate and extended family) and their wider communities. A recent qualitative report on rangatahi views of wellbeing in Aotearoa demonstrates the use of Te Whare Tapa Wha, wherein the Māori and Pasifika rangatahi referred to and used the framework to help understand their own wellbeing [98]. Research focusing on body image in Aotearoa has further illustrated that Māori and Pasifika draw upon their cultural ways of knowing to make meaning of their own and others' bodies, health, and wellbeing. For example, Houkamau and colleagues [99] conducted a large-scale study that documents how participation in traditional cultural practices that affirm cultural identity is strongly associated with higher levels of body satisfaction and self-esteem among members of the Māori population.

### 3. Methods

Mana Wahine and Masi methodology are the Indigenous feminist methodologies that we have used to frame this research. Mana Wahine magnifies the lived experiences and knowledge of Māori Wāhine, and it respects, protects, and centres their voices [100–102]. Similarly, Masi methodology centres and protects the voices and lived experiences of Pasifika wāhine [103–105]. We chose to weave Mana Wahine and Masi methodologies, as it is important that we acknowledge the cultural heritage of both Māori and Pasifika wāhine as well as those with mixed ethnicities. We resonate with recent scholarship that has used both Mana Wahine and Masi methodologies, in that we also "seek to honour both wāhine Māori and Pasifika in their own self-determination" [106] (p. 108).

This study draws upon the doctoral research of the first author, who conducted all of the data collection and led the analysis. Participants for this study were selected based on self-identifying as a wahine, being aged between 16 and 25 years old, having Māori and/or Pasifika ancestry, and reporting past or current participation in sport or physical activity. The young wāhine either currently lived, or had previously lived, in South Auckland (a group of suburbs in Auckland) and/or Porirua (a suburb in Wellington). South Auckland and Porirua were selected as locations for this study as they have similar demographics, including having a high Māori and Pasifika population, having a lower socio-economic status, and being highly urbanised regions within large cities.

In total, there were 31 young wāhine that were recruited using the current connections of the first author, through family, friends, social media contacts, and attendance at different group fitness venues. There were 16 young wāhine from South Auckland, and 15 from Porirua, and of these young wāhine, 17 identified as Māori, and 22 identified as Pasifika (Table 1). The majority of Pasifika wāhine were of Samoan descent, and 16 young wāhine had a mixed-ethnic background, with the most common being Māori–Samoan. Participants were given an information sheet, and they all voluntarily signed a consent form agreeing to be part of this study. The young wāhine were also given the opportunity to select a pseudonym, or have one assigned for them. Their self-selected or assigned pseudonyms are used throughout this article.

We purposefully selected young wāhine who were participating in sport or physical activity, as we wanted to take a strengths-based rather than deficit-based approach to our research. Too often, young Māori and Pacific wāhine are framed by media and policies that focus on their physical and social 'risks' and vulnerabilities, and we intentionally sought out alternative perspectives. In so doing, there is an Indigenous feminist politic underpinning this project. We acknowledge that the young women came to this project with varied past and present experiences of participating in sport and physical activity. Some were currently participating in competitive sports (i.e., netball, weightlifting, rugby); others had stopped participating in competitive sports, but valued physical activity in their everyday lives (i.e., walking, gyming). In this paper, we acknowledge the various ways in which their past and present sport and physical activity experiences may be shaping their experiences of body image, navigating various forms of body shame, and embracing body pride.

The varied characteristics among this cohort were typical for young women aged between 16 and 25 years in Aotearoa, whose lives are transitioning in many ways that can often be challenging. For example, this cohort of young wāhine ranged from being in high school to moving out of home, being full time mothers, or pursuing tertiary studies [107]. The range of participants for this research spans a range of mixed Māori and/or Pasifika ethnicities, and covers a varied range of ages and stages of life, and as such, provides a rich depth representative of the young wāhine whose voices we amplify in this research.

Two data collection methods were used in the broader study: (i) wānanga and individual interviews, and (ii) digital diaries. Two wānanga and individual interviews were conducted six to seven months apart. In the first wānanga, the young wāhine participated in semi-structured kōrerorero (discussions), after which they were given the opportunity to choose their preferred method to diarise their sport and fitness experiences. These experiences were discussed at the second wānanga. For young wāhine who were unable to make the scheduled wānanga, an individual interview was conducted at a time and place that best suited them. At each wānanga or interview, kai (food) was provided for participants and, where appropriate, karakia (prayer) was offered to begin the proceedings.

Recognising the importance of social media and digital technologies in the everyday lives of young wāhine, we drew inspiration from research wherein young wāhine were given the option to use digital methods to document their stories [12]. The digital method options recognise that many young people, including those in low socio-economic communities, use social media regularly, and view it as an integral part of their everyday lives [108,109]. Building upon and extending this work, we offered participants a range

of options to diarise their experiences. This practice aligned with the principles of Mana Wahine and Masi methodology, wherein participants are given full control of creating and recording their own stories [106]. The options suggested, and provided, included keeping a written journal (which was provided for them), participant-created videography and/or photography, shared social media posts (TikTok, Instagram, Facebook), and also the option to use a Google Form to record and document their stories. When the young wāhine asked whether they could use a mix of the options available, aligning with Mana Wahine and Masi methodologies, the first author wholeheartedly agreed. While the digital diaries were an important aspect of the overall research project, and key themes were identified across both methods, in this paper, we draw specifically from the wānanga and interviews.

**Table 1.** Participant Information.

| Pseudonym | Age | Ethnicity | Social Situation |
|---|---|---|---|
| | | South Auckland | |
| Deena | 18 | Māori | High school student. |
| Anne-Marie | 22 | Māori | Full time work; university graduate |
| Rayena | 23 | Māori | Full time work; mother of one |
| Evelyn Burns | 25 | Māori | Full time work; university graduate |
| Tati | 16 | Māori–Samoan | High school student |
| Hera | 17 | Māori–Samoan | Tertiary study |
| Vaitala | 21 | Māori–Samoan | Part time work |
| Lele | 22 | Māori–Samoan | Part time work; solo mother of one |
| Rangi | 24 | Māori–Samoan | Solo mother of two |
| Rita | 23 | Cook Island–Māori | Tertiary study |
| Lene | 20 | Cook Island | Tertiary study |
| Wenzie | 19 | Cook Island–Samoan | Tertiary study; part time work |
| Nora | 18 | Samoan–Niuean | High school student |
| Nadine | 24 | Samoan–Niuean | Tertiary study; part time work |
| Tee | 24 | Samoan | Tertiary study; part time work |
| Mary | 19 | Tongan | Tertiary study; part time work |
| | | Porirua | |
| Tanya | 19 | Māori | Full time work |
| Jean | 22 | Māori | Full time work; university graduate |
| Cherie | 23 | Māori | Tertiary study |
| Nina | 18 | Māori–Samoan | Tertiary study |
| Mischa | 25 | Māori–Samoan | Part time work; mother of one |
| Wairemana | 21 | Māori–Scottish–Swedish | Tertiary study |
| Charlotte | 24 | Māori–Pākeha | Full time work |
| Norah | 16 | Samoan | High school student |
| Sally | 17 | Samoan | High school student |
| Tali | 19 | Samoan | Full time work |
| Toni | 20 | Samoan | Full time work |
| Larma | 16 | Samoan–Tongan | High school student |
| Paige | 19 | Samoan–Tokelau | High school student |
| Mia | 20 | Samoan–Tokelau | Full time work |
| Rya | 23 | Samoan–Tokelau | Full time work |

We made extensive efforts to ensure that the young wāhine felt valued and appreciated for their time, and this was demonstrated from recruitment to the final wānanga. Some of the ways in which Mana Wahine and Masi methodologies were woven throughout the data gathering included nuanced cultural practices, such as opening wānanga with a karakia; providing more kai than needed, so that the young wāhine could make a plate of food to take home to their whanau (family); and a koha (gift or a donation), in the form of a t-shirt specially designed and printed by the first author. The Mana Wahine and Masi methodology values that were practiced and implemented throughout all engagements with the young wāhine included respect, reciprocity, and authentic connections [104,110]. Practices that were adapted to the young wāhine based on their stage of life and age, included being flexible with timeframes for the interviews and/or wānanga, and picking them up from, or dropping them off at, their homes.

It is important to note the positioning of the first author in relation to this research project, as she is Māori–Samoan, and has lived and worked in the South Auckland community for most of her life. Her diverse roles in South Auckland over the past few decades have included lecturer, gym owner, trainer, group fitness instructor, and world champion athlete. These roles have provided insights into the lived experiences of the young Māori and Pasifika wāhine in this research and many of them have been an important part of her life. While it would be remiss to assume she understands all cultural aspects of Māori and Pasifika, or that she shares the same experiences as the young wāhine in this research project, her embodied practices and lived experiences as a Māori–Samoan wahine in and from the South Auckland community provides nuanced insider perspectives.

## 4. Analysis: Amplifying the Voices of Young Māori and Pasifika Wāhine on Body-Image Ideals

*"How do you find beauty, when what is deemed beautiful, is never the reflection you see in front of you? How do you change the narrative when the stories that have been told are never the ones of your own people? Maybe it's time for a new voice. The dawning of a new day"*. [111]

This is a quote from a young Cook Island–Ghanian wāhine who is part of an Aotearoa-based online YouTube series through which young rangatahi with mixed ethnic identities share their lived experiences. We include it here, as it powerfully speaks to some of the themes in our research, and it highlights what we are aiming to achieve in this article, which is to share informed cultural knowledge of body image. In this section, we show how the young wāhine in our research are responding to dominant narratives of beauty and body image, and navigating alternative, culturally-informed ways of knowing their own bodies. In presenting our findings, we draw upon Mana Wahine and Masi methodology, which centres the voices of the young wāhine and their lived experiences, and we acknowledge them as knowledge creators, knowledge holders, and knowledge bearers [112,113]. Using a selection of quotes from wānanga, interviews and diaries, we amplify their voices across two different themes: (1) influences on body image, and (2) navigating body-image stereotypes.

### 4.1. Influences on Body Image

Interestingly, for the young wāhine, one of the biggest influences on their understanding of body image was connected to taha hinengaro, which is one of the pillars in the Te Whare Tapa Wha model [96]. Taha hinengaro is explained as being "your mind, heart, conscience, thoughts and feelings. It's about how you feel, as well as how you communicate and think" [114]. Rather than focusing on the aesthetics of looking a certain way, or having a particular body type or shape, the young wāhine felt that working on and strengthening the mental and spiritual aspects of their bodies was more important. This was an overwhelmingly shared belief among all of the young wāhine.

*People chase that image of looking good and having a six pack. I guess it's different for different people, but my approach to my health is looking after my mental state and it's spiritual things too. Like if I'm going to work out I'm working towards a goal, whether*

*it is rugby-related, it doesn't really have to be sports related to be honest. It's more working towards a better version of myself. I know that if I'm fit and well it just opens up so many opportunities because I'm ready, I'm prepared I know my body's prepared for opportunities.* (Nina, 18, Māori–Samoan)

*I think it's all about being mindset healthy and knowing when to put yourself first. Because for myself, I go out of my way a lot for other people, so knowing when to put my foot down and knowing when you deserve better. So, mindset-healthy kind of thing.* (Charlotte, 24, Māori–Pakeha)

*I think healthy looks when you're feeling more energised or looking better, glowing and feeling happy, and just looking after yourself more and having those positive thoughts about yourself, your body. You're not worrying about anyone else's opinion about your own body. I think healthy, like you want to change in your own way, not to fit someone else's type of body shape or what you need to look like as a woman or a girl. I think that's the best.* (Tali, 19, Samoan)

The young wāhine acknowledge the benefits and necessity of being physically active (i.e., going to the gym, doing sports) and good nutrition (i.e., eating vegetables). For example, Charlotte (24, Māori–Pākeha) identified health as "Eating the right vegetables and all that kind of stuff. Maybe going to the gym once a week" (Charlotte). Importantly, the reasons behind this behaviour are not related to body image in the Westernised sense (physical aesthetics), but rather to improving self-confidence and taha hinengaro. The young wāhine are embracing their cultural knowledge by sharing their views and beliefs of body image, and these align with body pride when they encourage confidence among the young women in embracing their bodies in a way that feels good to them [55]. Here, at the intersection of gender and ethnicity, we see a view of body image through the perspectives and experiences of the young Māori and Pasifika wāhine in this research.

Although the young wāhine expressed their beliefs about how body image and well-being relate to having good mental health, and taking care of their bodies through physical activity, they were also conscious of the influences of social media in their lives and those of the people around them.

*I think it's hard not to get influenced by social media, because hell, it's everywhere. Especially in this generation, social media is the number one thing...it influences the people that are growing up and that's what makes them kind of lose themselves in the process of their health, wellbeing and stuff like that.* (Mary, 19, Tongan)

Some of the young wāhine were affected by some of the images on social media, and commented on how they navigated these spaces:

*I used to buy into having a certain body type when I was Year 9 [roughly aged 13] on the insta-explore page. I really under-appreciated my body, but then as I got older it's really not realistic. I also noticed that a lot of models for some reason they were all white. I don't know why. I wish I had some black or some brown models on my explore page. That would have been nice. That would have influenced me. Mental health is just basically for me, it's not trying to get your mind sucked into all the social media stuff. It's all false reality, like you have to fit into a box. By leaving Instagram it helped me build up my confidence again and help my mental health to appreciate myself more.* (Norah, 16, Samoan)

*Sometimes I wish I could be like the Instagram body but at the same time I don't give a crap, I'm too lazy. I could if I would. But I don't have the means to at the moment. But at the same time, why do I want to look like that when I can just be myself? Just walk out with hairy legs? If no one can accept me for hairy legs then it's their problem. If I'm confident wearing hairy legs out, you know then, then cool. But like it's other people's problem, you know?* (Rangi, 24, Māori–Samoan)

In these quotes, we see the young wāhine critically analysing their own and others' experiences with images on social media. The young wāhine are actively problematising the lack of diversity on some social media platforms, and expressing a desire to see more

women that look like them. They also highlight the potential of such images to have a positive influence on young wāhine.

The young wāhine are also aware of the changing media landscape, and the influences of an ideal body image through the images on social media platforms. Recognising the growing diversity in images of women's bodies, they believe that society is starting to become more accepting of all body types:

> *In probably 2015, [the body shape you wanted to be like] was skinny, not bones but enough to feel around your whole waist or enough to fit into size small jeans and stuff, [that] is what you need to be. But now, people are accepting that not all body types are the same... But I think other people shouldn't have an opinion on women's body type, they shouldn't really have opinions on what we should look like, to be honest, because it's the body that we have.* (Tali, 19, Samoan)

Here, we see a young wahine actively critiquing dominant body-image discourses, and instead celebrating the multiplicity of bodies. She grounds herself in these beliefs by taking the power from people who have opinions about women's bodies and nullifying them, stating that young women's bodies belong to themselves, so they should have the freedom to interpret and present them however they choose. These statements illustrate body pride, as the comments of others that could prompt feelings of shame are rejected, and the voices and beliefs of the young wāhine are uplifted and valued [55].

The young wāhine shared critical reflections on how their perspectives on body image played out in daily interactions, and how these changed as they got older:

> *Looking around, I'd be like oh, this girl's got a boyfriend because she looks that type of way, or people like her because she's this type of body shape. That was my mentality of thinking of it when I was younger. But once I grew up, it was more about your personality. It's more important if you're nice, if you're kind, if you have all those attributes; it's not about your body type. I wish I thought about that when I was 15. But, when I was 18, I could see how that was negative thinking, because that's just bad. But to me, my ideal one would just be one that I'm comfortable in or one that I'm not shy to show off and stuff. If you're happy with that, then you do you.* (Tali, 19, Samoan)

In this example, we see development and learning in how te taha hinengaro began to increase in importance as Tali got older. She had a realisation that internal qualities, such as having a good personality and being kind, held more importance than body shape, and that her previous views when she was younger were negative. Internalising her views on body image, Tali then draws on body pride, in highlighting that not being shy, but being confident with how you feel about your body, and that being happy, whatever that might look like, is more important.

Being happy was a common narrative that was shared among the young wāhine, despite the negativity they sometimes face regarding having bigger bodies. Rita (23, Cook Island Māori) embodies the concept of being happy when sharing her response to a situation where she was told that she was getting "really fat." She responds jovially, "What do you mean I'm fat, I'm not! I'm just chubby' (laughing). Well, yeah, I'm happy. I'm happy the way I am." Despite being targeted with negative comments about how her body looks 'fat', she embraces how her body looks, and refuses to feel discouraged by these comments. Instead, she embraces the comment, then nullifies it by resolutely stating that she is happy with how she is. This also aligns with Teevale's [61] body-image research with Pacific youth, wherein one of the participants [a Tongan mother] believed that her daughter 'being happy' was more important than her being overweight.

Through an intersectional lens, we see that young Māori and Pasifika wāhine in this research presented views on body image that differ from research conducted on young White women [50]. Where the positive body-image characteristics in Wood-Barcalow et al. [50] referred to how they felt about the way their body looked in reference to the Western feminine ideals of being thin and fit, the young wāhine in this study shared views that were influenced by cultural knowledge shaped by te taha hinengaro.

*4.2. Navigating Body Image Stereotypes*

Many body-image stereotypes that are attributed to Māori and Pasifika include having larger bodies, and being physically dominant and 'naturally' athletic, particularly when compared to other ethnic groups in Aotearoa [50,115–117]. Indigenous scholars have argued against racialised ideologies relating to physical size and athleticism, as they limit the potential of Māori and Pasifika in other areas of society [116,118]. Having larger bodies also tends to frame Māori and Pasifika with having higher health risks, particularly with being overweight and/or obese, leading to higher rates of diabetes and cardiovascular health disease [119–122]. While we do not contest the health implications associated with obesity or being overweight, we argue that the socio-cultural stereotypes imposed on young Māori and Pasifika based on body size are unfair, and often unwarranted. Amplifying the voices of the young Māori and Pasifika wāhine in our research, we illustrate how they are navigating these 'big body' stereotypes, and enacting body pride, to powerfully claim their own bodies in public spaces and sport and fitness.

The young wāhine in our research were cognisant of the stereotypes relating to body size and were not hesitant in pointing these out. For example, Rita (23, Cook Island Māori) explained, "I think that's the way they look at us, Islanders are big people but to be honest, I just see them as human beings". Similarly, Mary (19, Tongan) reflected critically on stereotypes of Pasifika bodies: "Like they have big arms, yeah big arms, big bones, people look at them like, you shouldn't be that size, or like your arms shouldn't be like that. Sorry then God! (all laughing)". Here we see elements of body pride, as Rita (23, Cook Island Māori) rejects the notion of the stereotypes of Islanders having big bodies by ignoring size and focusing on people as humans. Mary (19, Tongan) also demonstrates body pride, and uses sarcasm and humour to contest the negative narratives placed on the larger bodies of Māori and Pasifika wāhine. Her sarcastic statement, "Sorry then God", is mocking people who dictate how female bodies should look. By using 'God', who is revered in many Pasifika cultures, and among some Māori [123,124], she is inadvertently stating that only 'God' has the power to dictate how things should be, and anyone else, such as those commenting on how Māori and Pasifika wāhine bodies 'shouldn't be', is overstepping their position in society. The idea and audacity of an individual behaving like 'God' is so preposterous that her comment incited laughter from the other young Māori and Pasifika wāhine in that particular wānanga. The laughter also demonstrated understanding, and that they agreed with this comment.

Such body-image pressures not only came from social media and wider society, but also within their own communities. Some of the young wāhine fielded negative body-shaming comments from friends and family, and navigated these situations in ways that protected themselves:

> *Earlier on this year, when I was back at the Islands, one of my cousins. . . called me fat. And I was like, that's not nice to say, but I actually take it into heart. And, yeah, and I felt like, Oh, I want to be skinny, I'm gonna start exercising, and then actually started exercising when I was in Raro. And I never missed a day of exercising, but when I was exercising I went back to my eating, like you know just eating anything I want to. . .the fatty food and stuff like that. And I say to myself, am I happy? And then I said, I don't think so and then I was like, you know what, I'm not gonna care if anyone says, 'You're fat' and things like that, I'm just gonna say, 'yeah nah, you can call me whatever you want but I don't care'.* (Rita, 23, Cook Island Māori)

> *In Samoa culture they take it just as a joke, "Hey, you're getting a bit chubbier there" or "You look a bit chubby there." It just defeats the purpose of love or is this what you're supposed to be hearing from your aunties or your uncles and stuff? When I was a little kid I was, "ha ha, that's funny." But now I'm grown up, you can see how that messes with you sometimes. I just laugh it off and I'm like, nah, I like the way I look now.* (Tali, 19, Samoan)

*I think I'm still the same really because I blocked it all out. Well, my dad used to always say I was skinny, [but] my Mum was like no, you're fine. Yeah, and I just happened to like put a guard up when anyone would say 'oh you're too small' or 'you're getting too big'. And yeah I would just like ignore it. Just because that's just how I am. . .I'll try and keep the negativity out because it's not good for your health.* (Nadine, 24, Samoan–Niuean)

Here, we see critical reflections from the young wāhine, wherein they recognised the negative views placed on them by family members (i.e., cousins, aunties and uncles, mothers and fathers), and chose to reject them, by drawing on their inner self-belief of what makes them happy. We see here that they are not immune to the negative comments, and that these seem to be ongoing. Notably, the young wāhine understood the detrimental effects that harmful body-image comments could have on their health, and consciously chose to 'block it all out', instead drawing on body pride to reject negative comments. Using phrases such as, "I don't care", "I like the way I look", and "I would just ignore it", they are confidently grounding themselves in their self-belief and self-confidence, despite having to navigate these comments.

Frustration was felt by some of the young wāhine with the negativity that came from family members or other Māori and/or Pasifika people. The following response from Mary (19, Tongan) expresses frustration and concern about negative body-image behaviour from people who are supposed to be supportive:

*Sometimes I want to stand up for that person cos we all come from the same place, same background, like we're all trying to make our parents proud and stuff. And like, that person may be having some problems at home and they come to school and they think coming to school as like a place where they can get their mind off the things that's happening at home. And just coming to school and having those people pick on you on your worst flaws. It just makes it even worse. And that's what leads some people to take their own life. I don't know like some people, they have the same background but they still have the audacity to come and do that to a person, like I just don't understand why you can't you just can't be nice to everyone, like my Mum has told me and my sisters, if you don't have anything nice to say don't say it at all. And just seeing how other people get treated because of just the way they look, how they weigh and like their size, like we're all, we're all not perfect, and we're all not made the same.*

This was a very emotive response that resulted in tears from everyone attending that particular wānanga. There was a shared and unspoken understanding of the pain and hurt caused by negativity from within one's own community, and the mention of suicide was a stark reminder of the damaging effects of such behaviour. Sadly, the ethnic groups with the highest rates of suicide in Aotearoa are Māori and Pacific people [125], so Mary's comment and the ensuing emotion indicated lived experiences relating to this. These lived experiences of collective feelings of body shaming from many parts of society, including within their own families and communities, may also explain why the young wāhine choose the characteristics of te taha hinengaro to reframe their understanding of body image in ways that protect their own mental health and wellbeing.

Using intersectionality, our analysis also revealed interesting insights into how young Māori and Pasifika wāhine come to understand their own gendered and ethnic bodies in sport and fitness contexts. While young Māori and Pasifika tāne (men) are lauded for having big, strong, muscular physiques that serve them well in sport and physical education spaces, young wāhine are often treated in a way that is quite the opposite. Herein, the young wāhine share their experiences of navigating sport and physical education spaces where their femininity is questioned, and their bodies become sites of critique under the gaze of a Westernised lens:

*They say it's like known to be ugly to be musclier than boys. Yeah, it's not feminine being muscle toned and all that. I used to get so pissed off and be like, 'oh, I'm gonna get skinny arms and all that', but it's never gonna happen (all laugh). You try but it's just like, no, it's not gonna happen, it's genetics, it's what you were born with and what God gave you.*

*Just continue being yourself. Just don't let anyone tell you any different. Just yeah from experience, listening to everyone's opinions, you know, I put my younger self down, so if I had a chance to tell [my younger self], I would say 'tell them to shut up and get outta my face'.* (Wenzie, 19, Cook Island–Samoan)

*Because my male PE teacher knew I did CrossFit and, you know, CrossFit was like, just coming out and everyone's like "It's not a sport", "It's just for men", "women, aren't supposed to lift". And I think that's what got me the most then I think like a few times I'd be smart and be like, well you have chicken legs (all laughing). My thighs can squash you (more laughing). And like just growing up doing weightlifting and CrossFit and just training was like the people that I was with you could just realize, well, I'm an Islander, this is how I was built and this is what I'm made for. So like yeah, I just think of it as like it's raw strength, use it while you can, while you're young. I'm like you have this body, use it while you can.* (Nadine, 24, Samoan–Niuean)

These experiences illustrate the negative attitudes young Māori and Pasifika wāhine encounter and negotiate surrounding the shape, size, and structure of their bodies. They described various instances of being discriminated against and treated unfairly because their bodies don't conform to Western standards of femininity and slim ideals. In the face of such stereotypes and treatment, however, these young wāhine powerfully claim and embrace their physical characteristics, by standing up to those who are judging them. They also use their bodies as sites of celebration of what they can do, rather than what they look like according to others' gendered and cultural expectations. The young wāhine argue against those who discriminate against them, by drawing on their own strengths ("My thighs can squash you"). Notably, Wenzie (19, Cook Island–Samoan) has recently accepted a contract to play for a prestigious Australian rugby league club, and Nadine (24, Samoan– Niuean) has represented Aotearoa internationally in weightlifting. It is important to point out that while these two young wāhine embrace their 'God given genetics' that have helped them to succeed in sport, and have used these successes to push back against cultural and gendered stereotypes, other wāhine who do not participate in sport and exercise may have different experiences and relationships with their bodies.

Another form of discrimination and stereotyping that the young wāhine had to navigate included experiences of being told they were too overweight to participate in sport:

*When I played reserve league for netball, we had to be a certain weight. We had to be 65 [kg] and under. You could never go above or else you'd get dropped. There are a lot of girls that deserved the spot more than other people, but just because of the weight restriction, they were cancelled out, which I reckon was pretty unfair. Some of the girls I've played with for years put in so much work and one girl just gets it because she's skinnier. It just changes the whole point of playing netball basically. Polynesians always had the weight restriction thing. We have bigger bodies, but big bodies can do a lot of things. [They] can do just as much as a little body can.* (Lene, 20, Cook Islander)

While Netball New Zealand policy guidelines identify exclusion based on weight as an unacceptable form of discrimination [126], our participants' experience suggest that such practices continue to occur, with negative consequences for Māori and Pasifika wāhine experiences in the sport. In defending the larger bodies of Pasifika wāhine, Lene (20, Cook Islander) rightly points out that "big bodies can do a lot of things".

Other young wāhine experienced weight-based stereotypes in school sports and physical education that negatively impacted their feelings about their bodies:

*Once she [teacher] saw me she was like, you need to lose weight and I actually did it because I was like 'okay, this is the sports coordinator, she's gonna make me the best athlete ever.' But then I got sick [from] doing that, so I stopped. But then just hearing all the opinions from everyone about me being you know too muscly, too big. At first it made me more angry than anything, the fact that they had the audacity to tell me that I'm overweight or too big for my age. But like, it's my genetics and I can't change that about myself, and I'm proud of that, of course!* (Wenzie, 19, Cook Island–Samoan)

Here, we see Wenzie experiencing the negative effects (anger, frustration) of such comments, but ultimately demonstrating refusal of others' judgements, and embracing body pride.

A few young wāhine expressed disbelief and frustration at some of the stereotypes they had experienced regarding the types of sport that they were 'expected' to participate in:

*At college (high school), most people wouldn't think Samoan girls would play netball. It's usually just, "I thought you played this kind of sport", like rugby and stuff because they think we're kind of like the Samoan boys. I reckon some of the girls that are Samoan have the ability to go just beyond playing at college. But I guess the other stereotype would just be that we'd do shotput and stuff, or big things that we can carry because we're of big stature and stuff. They've never seen people sprint, like in the 100 m one. I was like, oh my gosh, you've never seen a girl Samoan sprint, oh wow!* (Tali, 19, Samoan)

*Sometimes Islander girls are the stereotypes, like Islander girls should be playing rugby or contact, like men's sports. Not doing gymnastics and ballerina stuff. I get that one. If you're an Islander, doing gymnastics is like, oh you must be half something else. It's sad.* (Mary, 19, Tongan)

In these experiences, we see that young Māori and Pasifika wāhine are 'expected' to participate in particular types of sporting activities, based on their body types. Interestingly, some of the stereotyping comes from their peers, which could be seen as an indication of complicit beliefs. This peer stereotyping is problematic, as it can unconsciously limit participation in sports and activities that are outside of their scope, which, in turn places limits on their potential. Such findings resonate with the work of Cooky and Rauscher [127], who acknowledge that the participation of young women in sport is influenced by cultural factors, including expectations from others.

Showing resilience, and a deeply rooted belief in body pride, a few of the young wāhine drew strength from their sporting abilities to counter the stereotypes they faced relating to having bigger bodies:

*Being a rugby player it's, you know, you're muscley, you're, you're massive. I was, one of the biggest girls on my team. I'm not that big. Hey I'm not that big, (girls laugh reassuringly) but I was the biggest. I did reinforce my own you know attributes towards the game but then also, it's genetics like you know the muscliness, the strength. I'm gonna always have that from the stereotypes that all these people say about me. I'm not gonna, I can't hide it. I can't take my toughness away just for them, you know, so yeah I reinforce my stereotypes in a good way, like, kind of make bad into good, you know? I have my own individual strengths, like speed and stuff.* (Wenzie, 19, Cook Island–Samoan)

*A lot of white people, they just look at me and they're like 'oh, she's dumb, she doesn't know a lot'. Also, being a young woman, it also puts me under the bar as well because stereotypically we're weaker, meant to be weaker than males overall. Being both of those things is kind of hard because when people, mostly just white people, when they look at you it's kind of like they really underestimate you. They just look at you and think that you act a certain way. I don't like that. I guess in sport, sport really helps me to prove to people that I don't have to be a certain way. I can be really good at sport! For me, sport is the place where people can recognise you, see who you are as a person on the court or on the field.* (Norah, 16, Samoan)

As these comments suggest, for some Māori and Pasifika wāhine, sport is a place they can experience success and pride in what their bodies can achieve, and thus counter White, Western stereotypes based on gendered and racist ideologies. Despite the challenges and stereotypes faced by young wāhine, they are resolutely proclaiming their strengths and beliefs on body image, and demonstrating body pride, focusing on what their bodies can do.

Some young wāhine shared their views and recommendations on how they believed body-image views could be improved for young wāhine. Cherie (23, Māori) suggests, "I

definitely think pushing more of the confidence aspect of it. Because there will be bigger young girls who will just never look like that, [slim or slim-fit] but they can look great in their own way. So having that visual representation, pushing the confidence aspect, and pushing the community and healthy women aspects". Cherie leans into the concept of body pride through promoting self-confidence, while acknowledging that some young Māori and Pasifika wāhine may have bigger bodies that will never be able to fit Westernised body-image standards. Her recommendation of having visual representations of how young wāhine can 'look good in their own way' is important for building a community of healthy women, and working towards more nuanced knowledge and advice for body-image practitioners.

Summarising the voices of the young wāhine, Mary (19, Tongan) shares her final thoughts on how females should approach body *image:*

> *I think just getting everyone, especially females to understand that we're all given our own body type and we don't have to look like that person. It's not like a must, because I see it in social media there's a lot of girls that really want to hit a specific shape or type. I'm like, just be happy with your skin and your body, your body looks nice.*

Advocating body pride concepts by having an individualised appreciation of how their bodies look, Mary recommends that wāhine reject notions wherein body comparisons are made, and stop setting goals that involve striving to have a particular body shape. To all young wāhine who are having body-image concerns or struggles, she reassuringly states, 'be happy. . . your body looks nice'.

## 5. Conclusions

This paper contributes to a growing body of literature that demonstrates the importance of cultural identity in young people's understanding of health and wellbeing [94,128,129]. Working at the intersection of gender, cultural identity, and youth, our particular focus was on young Māori and Pasifika wāhine who are physically active in their everyday lives, and their understanding of body image. Drawing upon wānanga, individual interviews, and digital diaries involving 31 young Māori and Pasifika wāhine from two low socio-economic urban cities in Aotearoa, we reveal the ways in which young wāhine use cultural knowledge of taha hinengaro (which aligns with body pride) to challenge negative stereotypes, and to frame their understandings of body image. Although the young wāhine experienced negative stereotypes and body shaming in various aspects of their lives (i.e., school, sport), they draw on taha hinengaro and culturally-specific forms of body pride to embrace, and be happy with, their bodies, because their bodies belong to them.

In summary, our findings provide knowledge contextualised with, and for, young Māori and Pasifika wāhine in Aotearoa who are rejecting negative stereotypes about their bodies, and powerfully embracing how their bodies look and function. However, this article also highlights the need for those working with young Māori and Pasifika wāhine (i.e., teachers, coaches) to question some of their own assumptions, and critically reflect on how these may be negatively impacting the experiences of young Māori and Pasifika wāhine in a range of settings (i.e., school, sport). Some cultural stereotypes continue to abound and do harm. Furthermore, the dominant framings of young Māori and Pasifika bodies are out of date, and do not reflect the nuanced, culturally-informed strengths-based views that young wāhine have of their own bodies.

**Author Contributions:** Conceptualization, M.J.N.; methodology, M.J.N. and H.T.; formal analysis, M.J.N. and H.T.; investigation, M.J.N.; resources, M.J.N. and H.T.; data curation, M.J.N.; writing—original draft preparation, M.J.N.; writing—review and editing, M.J.N. and H.T.; supervision, H.T.; project administration, M.J.N. All authors have read and agreed to the published version of the manuscript.

**Funding:** This research received no external funding.

**Institutional Review Board Statement:** The study was approved by Te Manu Taiko: Human Research Ethics Committee in the Faculty of Māori and Indigenous Studies.

**Informed Consent Statement:** Informed consent was obtained from all subjects involved in the study.

**Data Availability Statement:** Some of the data presented in this study may be available on request from the corresponding author. The data are not publicly available due to ethical considerations and privacy of participants.

**Conflicts of Interest:** The authors declare no conflict of interest.

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
