# Peer review of "“My Thighs Can Squash You”: Young Māori and Pasifika Wāhine Celebration of Strong Brown Bodies"

_2673-995X, doi:10.3390/youth3030062_

Round 1

Reviewer 1 Report

The article has the merit of focusing on the body image knowledge for young women from indigenous and ethnic minority communities, particularly focusing on young Māori and Pasifika women, using intersectionality with culturally learned body positivity and body pride to reveal their ethnic and gender experiences. In particular aims to prevent increased risk of disordered eating practices and mental health problems resulting from strongly westernised body models part of the dominant narrative, especially prevalent in school and sports environments.

I think the article is well structured and the literature cited is adequate. However, with respect to the methodology where it states “in this paper we draw upon wānanga (focus groups) and digital diaries to examine how young Māori and Pasifika wāhine (16 to 25 years) who live in Aotearoa understand body image”, the results obtained from the diaries are not clear, which might be better emphasized in the analysis part. I therefore suggest stressing this part more in order to enhance the work done.

Reviewer 2 Report

This was a strong paper with interesting findings and the lead author obviously had good rapport with participants and generated good data as a result.

I have a few suggestions for the authors that I list below that I hope will attend to  few small details: 

1. Methodology - this is clear but there as one area which I felt left the paper a bit open. The participants have been selected from urban lower socio-economic suburbs in NZ - who all (?) attend group fitness venues. this is fine  but it becomes clear n the data that they don't necessarily represent all groups as at least Wenzie and Nadine are quite elite athletes/sports performers. the issue with this is that the 'bias' toward 'sporty' participants isn't stated  - who are likely to have quite a different body positivity to those who are less sporty (or at least worth noting). This then could shape the data and the article claims as it may have influenced attitudes towards bodies and sport and perceptions. 

2. Linked to this was a slight caution I felt in some statements that felt over-claiming (pg 11 ad 13) which noted that "we see a different view on body image for young Maori and Pasifika wahine when compared with previous research on NZ European women....' and again pg 13 'the wahine in this study were more concerned about holistic views' it is just a bit difficult from a small study to make such a claim as it might be the participants were a selection of participants that was biased to start with. I am just pointing this out as I felt it sounded like a bigger claim than the study might be able to justify given small numbers (and the possible 'sporty' bias? (as an aside the study no 50 only had 15 participants so was also not generalisable). 

3. The study made several statements about agency. Just caution that agency in sociological theory is not simple to see without understanding the context which it is developed within - so agency for one person may not be the same as for another (•White, R., & Wyn, J. (1998). Youth agency and social context. Journal of Sociology, 34(3), 314-327  )

4. Check whole ref list - lots of capitals for journal headings missing see ref 71, 105, 129 incomplete; Rangatahi spelt incorrectly pg 7. 
